# Antagonistic Roles of GRK2 and GRK5 in Cardiac Aldosterone Signaling Reveal GRK5-Mediated Cardioprotection via Mineralocorticoid Receptor Inhibition

**DOI:** 10.3390/ijms21082868

**Published:** 2020-04-20

**Authors:** Jennifer Maning, Katie A. McCrink, Celina M. Pollard, Victoria L. Desimine, Jennifer Ghandour, Arianna Perez, Natalie Cora, Krysten E. Ferraino, Barbara M. Parker, Ava R. Brill, Beatrix Aukszi, Anastasios Lymperopoulos

**Affiliations:** 1Laboratory for the Study of Neurohormonal Control of the Circulation, Department of Pharmaceutical Sciences, College of Pharmacy, Nova Southeastern University, Fort Lauderdale, FL 33328, USA; jm3706@mynsu.nova.edu (J.M.); km1911@mynsu.nova.edu (K.A.M.); cp1743@mynsu.nova.edu (C.M.P.); vd359@mynsu.nova.edu (V.L.D.); jg2901@mynsu.nova.edu (J.G.); ap2491@mynsu.nova.edu (A.P.); nc1174@mynsu.nova.edu (N.C.); kf713@mynsu.nova.edu (K.E.F.); barbaramparker@gmail.com (B.M.P.); avabrill@gmail.com (A.R.B.); 2Department of Chemistry and Physics, Halmos College of Natural Sciences and Oceanography, Nova Southeastern University, Fort Lauderdale, FL 33328, USA; ba285@nova.edu

**Keywords:** aldosterone, cardiac myocyte, G protein-coupled receptor kinase, G protein-coupled estrogen receptor, mineralocorticoid receptor, signal transduction

## Abstract

Aldosterone (Aldo), when overproduced, is a cardiotoxic hormone underlying heart failure and hypertension. Aldo exerts damaging effects via the mineralocorticoid receptor (MR) but also activates the antiapoptotic G protein-coupled estrogen receptor (GPER) in the heart. G protein-coupled receptor (GPCR)-kinase (GRK)-2 and -5 are the most abundant cardiac GRKs and phosphorylate GPCRs as well as non-GPCR substrates. Herein, we investigated whether they phosphorylate and regulate cardiac MR and GPER. To this end, we used the cardiomyocyte cell line H9c2 and adult rat ventricular myocytes (ARVMs), in which we manipulated GRK5 protein levels via clustered regularly interspaced short palindromic repeats (CRISPR)/Cas9 and GRK2 activity via pharmacological inhibition. We report that GRK5 phosphorylates and inhibits the cardiac MR whereas GRK2 phosphorylates and desensitizes GPER. In H9c2 cardiomyocytes, GRK5 interacts with and phosphorylates the MR upon β_2_-adrenergic receptor (AR) activation. In contrast, GRK2 opposes agonist-activated GPER signaling. Importantly, GRK5-dependent MR phosphorylation of the MR inhibits transcriptional activity, since aldosterone-induced gene transcription is markedly suppressed in GRK5-overexpressing cardiomyocytes. Conversely, GRK5 gene deletion augments cardiac MR transcriptional activity. β_2_AR-stimulated GRK5 phosphorylates and inhibits the MR also in ARVMs. Additionally, GRK5 is necessary for the protective effects of the MR antagonist drug eplerenone against Aldo-induced apoptosis and oxidative stress in ARVMs. In conclusion, GRK5 blocks the cardiotoxic MR-dependent effects of Aldo in the heart, whereas GRK2 may hinder beneficial effects of Aldo through GPER. Thus, cardiac GRK5 stimulation (e.g., via β_2_AR activation) might be of therapeutic value for heart disease treatment via boosting the efficacy of MR antagonists against Aldo-mediated cardiac injury.

## 1. Introduction

Aldosterone (Aldo) is among the hormones for which levels are elevated and contributes significantly to the pathogenesis and morbidity/mortality of heart disease, including hypertension and chronic heart failure (CHF) [1]. Most of its cardiotoxic effects are mediated by the mineralocorticoid receptor (MR), a transcription factor expressed in various tissues including the heart [2]. However, Aldo exerts MR-independent (“non-genomic”) actions, a great deal of which are through activation of the G protein-coupled receptor 30 (GPR30) or G protein-coupled estrogen receptor (GPER), a plasma membrane G protein-coupled receptor (GPCR) [3]. Most of the pathophysiological effects of Aldo in the heart and vessels, which can cause essential hypertension and trigger cardiac hypertrophy, fibrosis, and adverse remodeling, are thought to result from MR activation, whereas several studies suggest that GPER activation may actually be beneficial in the myocardium, partly due to activation of antiapoptotic mediators, such as Epidermal Growth Factor Receptor (EGFR) and extracellular signal-regulated kinase (ERK)1/2 [2,3,4,5,6,7]).

Most GPCRs undergo functional desensitization due to phosphorylation by GPCR-kinases (GRKs), a Ser/Thr protein kinase family, followed by binding of βarrestins which uncouple the receptor from its cognate G proteins [8]. GRK2 and -5 are the most abundant GRKs in the heart and are also known to phosphorylate non-GPCR substrates [9,10]. The MR is known to be phosphorylated at various Ser/Thr residues, including Ser-843 of the ligand-binding domain and Ser-601 of the N-terminal domain [11]. These phosphorylating events occur in the cytoplasm, and some of them result in MR inhibition either via direct repression of transcriptional activity (e.g., the Ser-843 phosphorylation) or via cytosolic retention, i.e., blockade of nuclear translocation (e.g., the Ser-601 phosphorylation) [11]. Herein, we sought to investigate the potential involvement of GRK2 and -5 in the regulation of cardiac Aldo signaling via the MR and/or GPER.

## 2. Results

### 2.1. β_2_AR-Activated GRK5 but Not GRK2 and Interacts with the MR in H9c2 Cardiomyocytes

Via co-immunoprecipitation (co-IP) experiments in cultured H9c2 cardiomyocytes, which endogenously express both β_1_- and β_2_-adrenergic receptors (ARs) [12] as well as the MR and GPER [13,14], we uncovered that GRK5 interacts with the MR constitutively and that this interaction is significantly boosted by β_2_AR activation with formoterol (Figure 1A,B). Of note, the non-subtype selective βAR agonist isoproterenol did not affect appreciably the GRK5–MR interaction (Figure 1A,B), suggesting that the β_1_AR might somehow compete with and offset the β_2_AR-mediated induction of this interaction in H9c2 cardiomyocytes. In contrast to GRK5, no GRK2 interaction with the MR could be detected under any treatment conditions (including constitutively, i.e., no cell treatment) in H9c2 cardiomyocytes (Figure 1A).

### 2.2. GRK2 but Not GRK5 Binds to and Desensitizes Agonist-Activated GPER in H9c2 Cardiomyocytes

Next, we checked for potential interactions of GRK2 and -5 with the plasma membrane-residing GPER in H9c2 myocytes. Co-IP experiments with GPER and the two GRKs revealed the exact inverse picture from the one with the MR above (Figure 1). Specifically, GRK2 was found to interact with GPER in an agonist-dependent manner (i.e., in response to either Aldo or the GPER synthetic agonist G1), whereas no GRK5 interaction with GPER could be detected in the presence or absence of either GPER agonist (Figure 2A).

Given that GRK2 interacts only with agonist-activated GPER, which is a GPCR, we speculated that the GRK2–GPER interaction probably results in classic desensitization of the receptor, i.e., G protein uncoupling. Indeed, this was confirmed via guanosine 5’-O-[gamma-thio]triphosphate, (GTPγS) experiments performed with GPER pretreated with G1 in the presence or absence of the known GRK2-specific small molecule inhibitor Cmpd101 [15] (Figure 2B). GRK2 blockade with Cmpd101 significantly increased the extent of G1-induced GTPγS binding of the G1-pretreated GPER (Figure 2B), a strong indication that GRK2 causes functional desensitization of GPER.

### 2.3. GRK5 Phosphorylates the MR, Inhibiting Its Transcriptional Activity in H9c2 Cardiomyocytes

We postulated that the MR is a phosphorylation substrate for the Ser/Thr kinase GRK5. Indeed, there are several Ser/Thr residues located within all three functional domains (N-terminal domain (NTD); central DNA-binding domain (DBD); and C-terminal ligand-binding domain (LBD)) of the human MR protein (UniProtKB #P08235) [11] that are excellent substrates for GRK5-mediated phosphorylation, as predicted by the NetPhos 2.0 Protein Phosphorylation Prediction Server software (http://www.cbs.dtu.dk/services/NetPhos/np.html) [16]. Using a general anti-phosphoSer antibody to measure the phosphorylated Ser content of immunoprecipitated MR in H9c2 cardiomyocytes overexpressing GRK5 (via transfection with a lentivirus encoding for full length GRK5), we found that the phospho-Ser content of the MR was markedly augmented in GRK5-overexperssing cardiomyocytes compared to control, mock (empty vector) virus-transfected cells (Figure 3A,B). Conversely, in cardiomyocytes lacking GRK5 due to lentivirus-mediated clustered regularly interspaced short palindromic repeats (CRISPR)/Cas9 rat GRK5 gene deletion (Figure 3C), the phospho-Ser content of the MR was strikingly diminished compared to control, mock CRISPR virus-transfected cells (Figure 3A,B). Interestingly, the MR phosphorylation detected in these experiments was unaffected by the presence or absence of Aldo (Figure 3A,B), similarly to the GRK5-MR interaction above (Figure 1). Finally, in vitro kinase assay experiments with human recombinant GRK5 and resin-immobilized human recombinant MR protein confirmed that the MR is directly phosphorylated by GRK5. Taken together, these results strongly suggest that the cardiac MR is a phosphorylation substrate of GRK5.

Next, we examined the impact of GRK5-induced phosphorylation on the transcriptional activity of the MR in H9c2 cardiomyocytes. Employing the luciferase gene reporter assay in cells expressing the luciferase gene under the influence of the MR gene (*NR3C2*) promoter, we found that Aldo-dependent MR transcriptional activity was significantly enhanced in GRK5-depleted cardiomyocytes but virtually abrogated in GRK5-overexpressing H9c2 cardiomyocytes (Figure 3D). This strongly indicates that GRK5-induced phosphorylation of the cardiac MR leads to inhibition of its Aldo-dependent transcriptional activity.

### 2.4. GRK5 Protects Against Aldo-Induced Apoptosis/Oxidative Stress and Is Necessary for Eplerenone’s Inhibitory Actions in ARVMs

Since the cardiotoxic effects of Aldo, including apoptosis, oxidative stress, fibrosis, and inflammation, are almost exclusively mediated by the MR [2,5], the finding that GRK5 phosphorylates and inhibits the cardiac MR hints at potential cardioprotective effects of this kinase. Thus, we measured the effects of GRK5 on Aldo signaling also in cultured adult rat ventricular myocytes (ARVMs), a bona fide cardiomyocyte cell system even more physiologically relevant than H9c2 myocytes. We also tested the effects of GRK5 in ARVMs in the presence or absence of the known MR antagonist (MRA) drug eplerenone [17]. As shown in Figure 4A,B, eplerenone was only partially effective at blocking Aldo-induced apoptosis in control ARVMs. In GRK5-overexpressing ARVMs however (Figure 4C), aldosterone’s efficiency at promoting apoptosis was significantly reduced compared to control ARVMs and pretreatment with eplerenone resulted in complete abrogation of Aldo-induced apoptosis (Figure 4A,B). Conversely, in ARVMs having GRK5 deleted via CRISPR/Cas9 (Figure 4C), eplerenone was essentially incapable of blocking Aldo-stimulated apoptosis (Figure 4A,B). Similar results were obtained when Aldo-dependent oxidative stress generation was measured in ARVMs (Figure 4D). Indeed, GRK5 overexpression partially blocked Aldo-stimulated reactive oxygen species (ROS) generation and enabled eplerenone to abolish it completely in ARVMs (Figure 4D), whereas, in the absence of GRK5, this MRA was incapable of preventing Aldo-stimulated ROS generation in ARVMs (Figure 4D). Taken together, these results indicate (a) that GRK5 inhibits Aldo-dependent cardiac apoptosis and oxidative stress and (b) that GRK5 is essential for eplerenone’s cardioprotective actions against these deleterious effects of Aldo in the myocardium.

### 2.5. MR Inhibition by GRK5 in ARVMs Is β_2_AR-Inducible

Our results from H9c2 cardiomyocytes indicated that β_2_AR activation augments the interaction between GRK5 and the cardiac MR. On the other hand, GRK5 is activated by the β_2_AR (itself a substrate of this kinase) [18]. Therefore, we sought to interrogate further the role of the β_2_AR in GRK5-dependent MR inhibition in cultured ARVMs. As shown in Figure 5A,B, β_2_AR-selective activation with salbutamol but not β_1_AR-selective activation with dobutamine significantly increased the (Ser) phosphorylation of the endogenous MR. Of note, neither Aldo nor isoproterenol had any appreciable effect on MR phosphorylation in ARVMs (Figure 5A,B), similarly to what was observed with the GRK5–MR interaction in H9c2 cells (Figure 1). Importantly, β_2_AR-induced MR phosphorylation was GRK5-mediated, since transient transfection of the ARVMs with a kinase-dead GRK5 (K215R) dominant negative mutant [19] markedly suppressed basal and completely abrogated salbutamol-stimulated MR phosphorylation in ARVMs (Figure 5C,D).

To confirm that β_2_AR-induced, GRK5-mediated MR phosphorylation leads to inhibition of transcriptional activity also in ARVMs, we measured Aldo-dependent mRNA induction of the plasminogen activator inhibitor (PAI)-1 gene, one of the immediate/early Aldo-inducible genes in target tissues [20,21]. Indeed, β_2_AR activation with salbutamol completely blocked Aldo-induced PAI-1 mRNA expression in ARVMs (Figure 5E). Importantly, this inhibitory effect of the β_2_AR was GRK5-dependent, since salbutamol had no effect on Aldo-induced PAI-1 mRNA upregulation in ARVMs transfected with the kinase-dead GRK5 mutant (Figure 5E). Taken together, these results strongly suggest that the cardiac β_2_AR (but not the β_1_AR) induces GRK5-mediated phosphorylation of the cardiac MR, thereby inhibiting the latter‘s transcriptional activity.

### 2.6. Cardiac β_2_AR-Stimulated GRK5-MR Interaction Is Cytoplasmic and Ca^2+^-Calmodulin (CaM)-Dependent

To delineate the signaling mechanism(s) underlying stimulation of GRK5-mediated phosphorylation of the cardiac MR by the β_2_AR, we first examined the subcellular localization of the GRK5–MR interaction in H9c2 cardiac myocytes. Since both the MR and GRK5 can localize in both the cytoplasm and the nucleus [5,22,23], one important question arising from our present findings is which subcellular fraction these two proteins interact with each other in. To this end, we subjected H9c2 cell lysates to subcellular fractionation and we performed co-IP experiments in cytosolic and nuclear fraction extracts. As shown in Figure 6A, the GRK5–MR complex could be detected only in the cytosolic fraction extracts, both in the absence and presence of Aldo. Additionally, overexpression of a dominant negative GRK5 mutant permanently “trapped” in the nucleus due to lack of a nuclear export sequence (ΔNES-GRK5) [23] was incapable of interacting with the MR in H9c2 whole cell lysates. Therefore, GRK5 interacts with and phosphorylates the MR in the cytoplasm, causing retention of the Aldo–MR complex in the cytosol, i.e., blockade of the complex‘s gene transcriptional activity.

Given that GRK5 is normally plasma membrane-anchored, thanks to its constitutive interaction with membrane phospholipids [9], an important question pertains to the mechanism of its translocation to the cytoplasm, wherein it phosphorylates the MR. Recent studies have revealed that the ubiquitous Ca^2+^-binding protein CaM, known to inhibit GRK5 membrane localization and activity toward GPCR phosphorylation [24], can markedly increase GRK5 activity towards non-GPCR (noncanonical) substrates as well as can induce GRK5 translocation to the nucleus [22]. The latter process has been reported to mediate GRK5-depedendent cardiac hypertrophy [23]. Therefore, we hypothesized that Ca^2+^ signaling through CaM activation may be involved in the β_2_AR-stimulated GRK5–MR interaction in the cytosol of H9c2 cardiomyocytes. Indeed, pretreatment of cells with the phospholipase C (PLC)-β inhibitor U73122 [25] abolished the salbutamol-enhanced (but not the basal) GRK5 interaction with the MR in H9c2 whole lysates (Figure 6B,C). Since PLCβ activation induces Ca^2+^ signaling and activates CaM via Ca^2+^ release from intracellular 1’, 4’, 5’-inositol trisphosphate (IP_3_)-gated stores [26], this result suggests that β_2_AR-activated GRK5 translocates to the cytosol to phosphorylate the MR in a PLCβ-Ca^2+^-CaM-dependent manner.

## 3. Discussion

In the present study, we have uncovered a new, noncanonical, and advantageous function of cardiac GRK5: inhibitory phosphorylation of the MR (Figure 7). This GRK5 effect is β_2_AR-dependent and proceeds through a noncanonical PLCβ-Ca^2+^-CaM signaling pathway (Figure 7), resulting in GRK5 translocation from the plasma membrane to the cytoplasm [22]. The MR has long been established as an important molecular culprit in heart disease progression [27]. In fact, the MR (but not the very closely related glucocorticoid receptor) was recently shown to promote cardiac dysfunction/cardiomyopathy, even without cardiac insult or injury, such as infarction or aortic constriction, in transgenic mice [28]. This suggests that all of the MR-dependent cardiac effects of Aldo are harmful whereas the effects of glucocorticoids, such as the endogenous glucocorticoid hormone cortisol, may actually be beneficial or protective for the myocardium. Indeed, there is vast array of studies over the past 15 years or so confirming the deleterious role of the MR specifically in the heart and laying the mechanistic foundation for the use of MRA drugs for the treatment of advanced human CHF [1,2,5,27].

The MR undergoes various posttranslational modifications, such as phosphorylation, ubiquitination, etc., which regulate its transcriptional activity and/or ligand binding specificity/affinity [11,29]. Phosphorylation of cofactors required for MR transcriptional activity plays a role as well [30]. GRKs are known to phosphorylate not only agonist-activated GPCRs in order to desensitize them but also non-GPCR substrates [9,23]. GRK2 and GRK5 are the most abundant isoforms in extraretinal tissues, and GRK2, cytoplasmic when inactive, requires interaction with the free Gβγ subunits of activated G proteins for activation/membrane translocation [9]. In contrast, GRK5 is ionically bound to cell membrane phospholipids via a highly basic region of its C-terminus, so it is usually cell membrane-anchored [31]. Notably, the MR was recently shown to promote heart failure by activating GRK2-dependent apoptosis and GRK5 nuclear accumulation-dependent hypertrophy in transgenic mouse hearts in vivo [30]. These noncanonical, deleterious effects of the two GRKs were mediated by MR-induced, c-Src kinase-dependent cardiac angiotensin II type 1 receptor (AT_1_R) transactivation [22,30].

In the present study, we have uncovered that these two GRKs can also regulate Aldo receptor signaling in cardiac myocytes in their own right. Specifically, GRK5 (but not GRK2) phosphorylates the MR in both H9c2 cardiomyocytes and in ARVMs, inhibiting its transcriptional activity, while GRK2 phosphorylates and desensitizes GPER (Figure 7). Moreover, this noncanonical effect of GRK5 is β_2_AR-dependent. Moreover, it appears essential for the cardioprotection afforded by MRA drugs like eplerenone against Aldo’s deleterious effects. Nevertheless, the precise effects of GRK5-dependent MR phosphorylation on eplerenone’s MR inhibitory efficacy warrant further investigation and will be the focus of our future studies. In addition, identification of the exact Ser/Thr residues of the MR phosphorylated by GRK5 is already under way in our laboratory. An important question that arises is why GRK2 appears incapable of phosphorylating the MR as GRK5 does. This awaits further investigation. However, GRK5 phosphorylates the MR in the cytoplasm in order to block nuclear translocation and subsequent gene transcription activation, and GRK2, when activated, is located in the plasma membrane [9,23]. Thus, it is tempting to speculate that the reason might be simply the different subcellular compartmentalization of active GRK2 and the MR that prevents their interaction. Of note, GRK5’s nuclear/genomic effects, as a class II histone deacetylase (HDAC) kinase, have been postulated to be harmful (i.e., pro-hypertrophic) in the heart [23]. Therefore, the subcellular localization of GRK5 seems to be of paramount importance for this kinase‘s cardiac effects: inside the nucleus, it promotes maladaptive hypertrophy, but in the cytoplasm, it has beneficial effects via inhibition of the MR. Cardiac MR blockade is another line of evidence for GRK5’s overall beneficial, protective role in the myocardium. Indeed, enhanced GRK5 activity has been associated with favorable, β-blocker treatment-like outcomes in human CHF [32] with attenuated atherosclerosis [33] and with better cardiac function in humans/CHF patients [34]. Additionally, GRK5 has been well documented to inhibit cardiac nuclear factor (NF)-κB, resulting in reduced myocardial inflammation [35,36]. In contrast, every cardiac effect of GRK2 reported thus far appears deleterious for cardiac function and/or structure [8,26], and our present study adds one more to this list: cardiac GPER desensitization.

GPER (or GPR30) is a GPCR responsible for some of the rapid (non-genomic) effects of estrogen [4,6,7]. It is expressed in both cardiac myocytes and fibroblasts and can couple to both G_i_ and G_s_ proteins inducing EGFR or insulin-like growth factor-1 receptor (IGF-1R) transactivation in various tissues [4,6,7]. The precise cardiovascular effects of GPER activation in the heart and in vessels are still under intense investigation, but the majority of the studies indicate beneficial roles for this receptor, such as vasodilation; antiapoptosis/proliferation of vascular smooth muscle and endothelial cells; high-salt-induced diastolic dysfunction attenuation; reduction in left ventricular filling pressure, mass, and wall thickness; improved cardiac function; and inhibition of cardiac fibroblast proliferation [6,7]. Therefore, the fact that GRK2 opposes the antiapoptotic GPER signaling in cardiac cells is consistent with this kinase’s well-established role in cardiac apoptosis and may represent another mechanism underlying the cardiotoxic effects of this kinase [23].

As far as the signaling pathway that underlies the stimulation of GRK5 by the β_2_AR to phosphorylate the cardiac MR in the cytoplasm is concerned, it is interesting to note that the β_1_AR does not seem to share this ability. In fact, it probably opposes this GRK5 effect, since isoproterenol, which activates both β_2_- and β_1_ARs equipotently, does not promote GRK5–MR association (Figure 1) or MR phosphorylation by GRK5 (Figure 5A,B). The reason for this will be the focus of future investigations. The obvious explanation would be that the β_1_AR cannot activate the PLCβ-Ca^2+^-CaM pathway as the β_2_AR can and, thus, cannot induce the “shedding” of GRK5 from the cell membrane. In support of this, a recent study reported that protein kinase A (PKA), robustly activated by the cardiac β_1_AR, inhibits β_2_AR’s ability to stimulate Ca^2+^-CaM signaling via ERKs and IP_3_ in the caveolae of atrial myocytes [37]. It is thus quite plausible that the β_1_AR antagonizes the β_2_AR-induced, Ca^2+^-dependent GRK5 cytoplasmic/nuclear translocation in cardiac myocytes [22]. Of note, although it normally does not couple to G_q_ proteins, the β_2_AR has been reported to directly activate PLCβ, eliciting intracellular IP_3_-dependent Ca^2+^ mobilization/signaling [38].

## 4. Materials and Methods

All drugs/chemicals were from Sigma-Aldrich (St. Louis, MO, USA), except for Cmpd101 (HelloBio, Cat. #HB2840, Bristol, UK) and U73122 (MilliporeSigma, Cat. #CAS 112648-68-7, Burlington, MA, USA).

### 4.1. Cell Culture, Viruses, and Transfections

The H9c2 rat cardiomyoblast cell line was purchased from American Type Culture Collection (Manassas, VA, USA) and cultured as previously described [12,39]. ARVMs were isolated from adult male Wistar-Kyoto rats and cultured, as previously described [40]. H9c2 cells were used because they are a universally accepted and widely used cell model system for signaling studies in cardiac myocytes. Nevertheless, they are not bona fide cardiac myocytes (for instance, they do not contract); for this reason, ARVMs, which are bona fide, fully differentiated cardiac myocytes, were also used. Recombinant lentiviruses encoding for human wild-type full-length GRK5 or for empty vector (control) (OriGene Technologies, Rockville, MD, USA) were propagated and purified via CsCl density gradient ultracentrifugation, as described previously [25,39]. For CRISPR/Cas9-mediated GRK5 gene deletion, a gRNA sequence was custom-synthesized by Sigma-Aldrich (target ID: RN0000391809, target sequence: 5’-GTGGTTTGAATTTATGCGG-3’, predicted efficiency: 69.8, predicted specificity: 99.7) and incorporated into a lentiviral vector (Sigma-Aldrich). Along with negative control CRISPR lentiviral particles (CNCV, Cat #CRISPR12V-1EA, Sigma-Aldrich), this lentivirus was also propagated and purified through cesium chloride density gradient ultracentrifugation. The kinase-dead GRK5 (K215R) [19] mutant cDNA was generated using the QuikChange site-directed mutagenesis kit (Agilent Technologies, Santa Clara, CA, USA), and proper insertion of the point mutation was verified by PCR. The mutant cDNA was then inserted into a lentiviral vector for lentiviral particle production and packaging (custom-made by Vigene Biosciences, Rockville, MD, USA).

### 4.2. Co-Immunoprecipitation (Co-IP) and Western Blotting

H9c2 or ARVM cell extracts were prepared, as described previously [39], in a 20-mM Tris pH 7.4 buffer containing 137 mM NaCl, 1% Nonidet P-40, 20% glycerol, 10 mM phenylmethylsulfonylfluoride (PMSF), 1 mM Na_3_VO_4_, 10 mM NaF, 2.5 µg/mL aprotinin, and 2.5 µg/mL leupeptin. Protein concentration was determined (Pierce BCA Protein Assay Kit, Thermo Scientific, Waltham, MA, USA), and equal amounts of protein per sample were used for IP or western blotting. MR was immunoprecipitated by overnight incubation of extracts with an anti-MR antibody (#ab62532; Abcam, Cambridge, MA, USA), attached to Protein A/G-Sepharose beads (Sigma-Aldrich), while GPER was immunoprecipitated with an anti-GPER antibody ((#ab39742; Abcam), also attached to Protein A/G-Sepharose beads (Sigma-Aldrich). The IPs were then subjected to immunoblotting for GRK2 (sc-562; Santa Cruz Biotechnology, Santa Cruz, CA, USA) or GRK5 (sc-565; Santa Cruz Biotechnology) or for phosphoserine (AB1603; Millipore-Sigma) to measure the pSer content of the immunoprecipitated MR. For the subcellular fractionation of H9c2 cells, cells were lysed using the NE-PER Nuclear and Cytoplasmic Extraction Reagents (Cat. #78833; Thermo-Scientific, Rockford, IL, USA), according to the manufacturer’s instructions. Primary antibodies to detect tubulin (#ab56676; Abcam), as a cytoplasmic fraction marker, and lamin A (#ab108922; Abcam), as a nuclear fraction marker, were used. Finally, an anti-glyceraldehyde 3-phosphate dehydrogenase (GAPDH) antibody (sc-25778; Santa Cruz Biotechnology) was used to control for protein loading. All immunoblots were revealed by enhanced chemiluminescence (ECL, Life Technologies, Grand Island, NY, USA) and visualized in the FluorChem E Digital Darkroom (Protein Simple, San Jose, CA, USA), as described previously [39].

### 4.3. Luciferase Reporter Activity Assay

Luciferase reporter activity assay was performed by transfecting the cells with the LightSwitch™ luciferase reporter gene vector under the influence of the MR promoter (Active Motif, Inc., Carlsbad, CA, USA) [41]. The measurements were done the next day with the manufacturer’s LightSwitch™ assay kit and according to its instructions.

### 4.4. TUNEL and Real-Time PCR

Terminal deoxynucleotidyl transferase-mediated dUTP nick-end labeling (TUNEL) assay to measure apoptotic cell death was done as described [42]. Briefly, cells were fixed with 10% neutral buffered formalin, embedded in paraffin, and sectioned at 5-µm thickness. DNA fragmentation was detected in situ in deparaffinized sections using the ApopTag peroxidase in situ apoptosis detection Kit (EMD Millipore) and according to manufacturer’s instructions. The total number of nuclei was determined by manual counting of 4′,6-diamidino-2-phenylindole (DAPI)-stained nuclei in six random fields per section. All terminal deoxynucleotidyl transferase-mediated dUTP nick end-labeling (TUNEL)-positive nuclei were counted in each section. Real-time PCR for rat PAI-1 mRNA levels in total RNA isolated from ARVMs was done as described previously [39,42,43]. Briefly, quantitative real-time PCR was performed using a MyIQ Single-Color Real-Time PCR detection system (Bio-Rad Laboratories, Hercules, CA, USA) using SYBR Green Supermix (Bio-Rad) and 100 nM of gene-specific oligonucleotides. Quantification of mRNA included normalization to 18s rRNA levels. No bands were seen in control reactions in the absence of reverse transcriptase. Primer pairs used were 5’-TTCCTCCACAGCCATTCTAGTCT-3’ and 5’-GAAAGGATCGGTCTAAAACCATCTC-3’ for PAI-1; 5′-TCGATGCTCTTAGCTGAGTG-3′ and 5′-TGATCGTCTTCGAACCTCC-3′ for 18S rRNA.

### 4.5. Oxidative Stress Assay

To determine reactive oxygen species (ROS) production, the 2′,7′-dichlorofluorescein diacetate (DCFDA) dye-based assay kit from Molecular Probes (Cat. #C13293; Eugene, OR, USA) was used, and the measurements were done according to manufacturer’s instructions and as previously described [44]. Briefly, cell extracts were incubated with 2 µM DCFDA for 20 min and ROS production was monitored by determining the fluorescence intensity using a fluorescent plate reader in which excitation and emission wavelengths were set at 495 and 520 nm, respectively. The fluorescence OD values obtained were normalized with protein determination and expressed as % of the values obtained in vehicle (1 µM DMSO)-treated cells.

### 4.6. GTPγS Assay

The GTPγS assay was performed as we have previously described [45]. Briefly, cell membranes were prepared by centrifugation and resuspension in hypotonic lysis buffer. The reaction was started by adding an aliquot of membrane suspension to reaction buffer containing 25 mM Tris-HCl, pH 7.4, 5 mM MgCl_2_, 1 mM ethylenediaminetetraacetic acid (EDTA), 1 mM dithiothreitol, 100 mM NaCl, 1 μM GDP, and 2 nM [^35^S]-GTPγS (from PerkinElmer, Billerica, MA, USA) with or without agonist. Reaction was stopped with ice-cold Tris-HCl buffer, pH 7.4, followed by filtration (Whatman GF/B glass fiber filters, MilliporeSigma) and scintillation counting. Ten μM (cold) GTPγS was used to measure nonspecific binding. Cells were pretreated with 100 nM G1 in the presence or absence of 30 μM Cmpd101, followed by a 100 nM G1 challenge for 15 min in order to determine agonist-induced desensitization.

### 4.7. Statistical Analysis

Student’s *t* test and one- or two-way ANOVA with Bonferroni test were used for statistical comparisons, unless otherwise indicated. For multiple group analyses, Dunnett’s test with SAS version 9 software (Cary, NC, USA) was also used. A *p* value of <0.05 indicated statistical significance.

## 5. Conclusions

We have uncovered a new unconventional (non-GPCR) substrate for cardiac GRK5, the MR, which results in cardioprotective effects of this kinase against Aldo. In contrast, GRK2 phosphorylates and desensitizes cardiac GPER with potentially cardiotoxic consequences. Admittedly, the major limitation of our study is that its findings await in vivo validation. Nevertheless, we have confirmed this novel role for cardiac GRK5 in a bona fide cardiomyocyte cell system (ARVMs) and we have delineated the underlying signaling mechanism. Upon confirmation in vivo, these findings may provide additional evidence for the emerging role of GRK5 as the GRK isoform acting as the protective ballast against the generally deleterious GRK2 in the heart. Finally, GRK5-dependent MR inhibition appears essential for the cardioprotection afforded by MRA drugs, like eplerenone, against Aldo. Given the problems and limitations hampering clinical efficacy of this drug class for advanced human CHF and other heart diseases [5], cardiac GRK5 stimulation (either directly with a small organic molecule or indirectly via β_2_AR-selective agonism) may offer a new pharmacotherapeutic avenue to augment the cardiac benefits of Aldo inhibitor drugs.

## Figures and Tables

**Figure 1 ijms-21-02868-f001:**
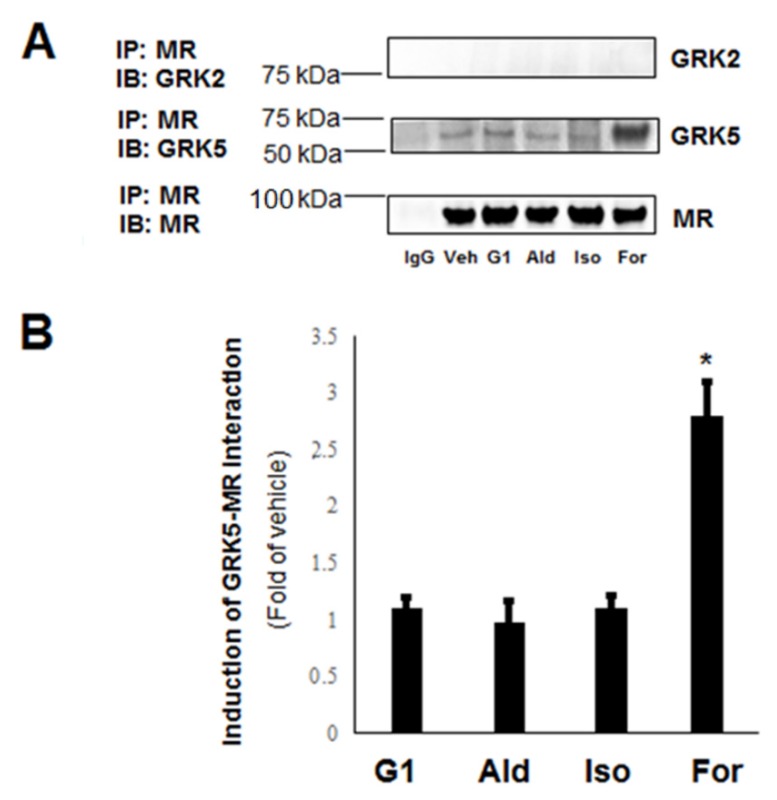
Interaction of G protein-coupled receptor (GPCR)-kinase (GRK)-5 but not GRK2 with the mineralocorticoid receptor (MR) in cardiac myocytes and the role of β-adrenergic receptors (ARs): (**A**) Immunoblotting for GRK2 and GRK5 in MR immunoprecipitates from H9c2 cardiomyocyte extracts prepared after 15-min-long treatments with vehicle (dimethylsulfoxide—DMSO, Veh), 100 nM G1, 100 nM aldosterone (Ald), 10 µM isoproterenol (Iso), or 10 µM formoterol (For). Blots for MR are also shown to confirm equal amounts of receptor immunoprecipitated. IP: immunoprecipitation; IB: Immunoblotting; IgG: Negative control for the co-IP (general rabbit IgG was used in the IP instead of an anti-MR antibody). Representative blots are shown in (**A**) and the relative densitometric quantitation of five independent experiments done in duplicate is shown in (**B**). * *p* < 0.05, vs. any other treatment; *n* = 5.

**Figure 2 ijms-21-02868-f002:**
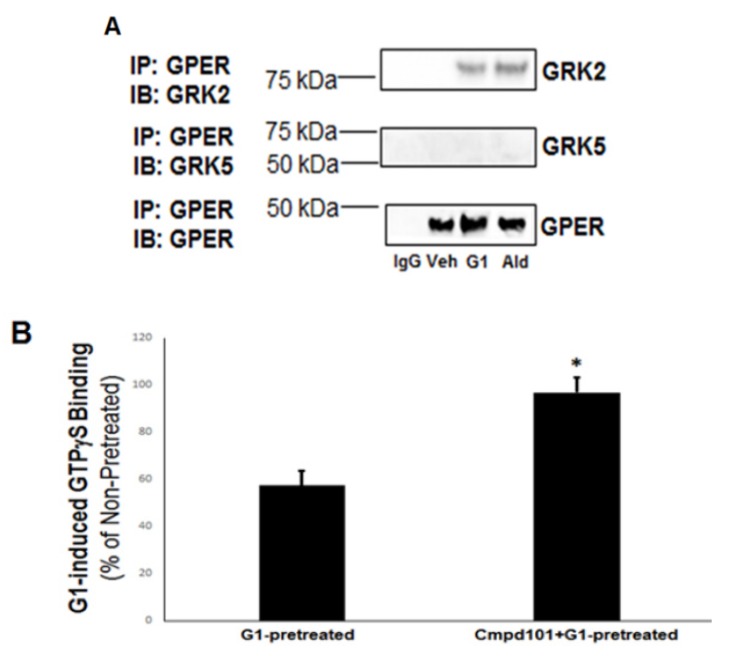
GRK2 but not GRK5 desensitizes G protein-coupled estrogen receptor (GPER) in cardiac myocytes. (**A**) Immunoblotting for GRK2 and GRK5 in GPER immunoprecipitates from H9c2 cardiomyocyte extracts prepared after 15-min-long treatments with vehicle (DMSO, Veh), 100 nM G1, or 100 nM aldosterone (Ald). Blots for GPER are also shown to confirm equal amounts of receptor immunoprecipitated. IP: immunoprecipitation; IB: Immunoblotting; IgG: Negative control for the co-IP (general rabbit IgG was used in the IP instead of an anti-GPER antibody). (**B**) Total GTPγS binding to measure G1-induced GPER desensitization in H9c2 cardioymyocytes: Cells were treated with 100 nM G1 for 15 min following a 30-min-long pretreatment with either 100 nM G1 alone (G1-pretreated) or 100 nM G1 in the presence of 30 µM Cmpd101 (Cmpd101 + G1-pretreated). *, *p* < 0.05; *n* = 5 independent experiments performed in triplicate.

**Figure 3 ijms-21-02868-f003:**
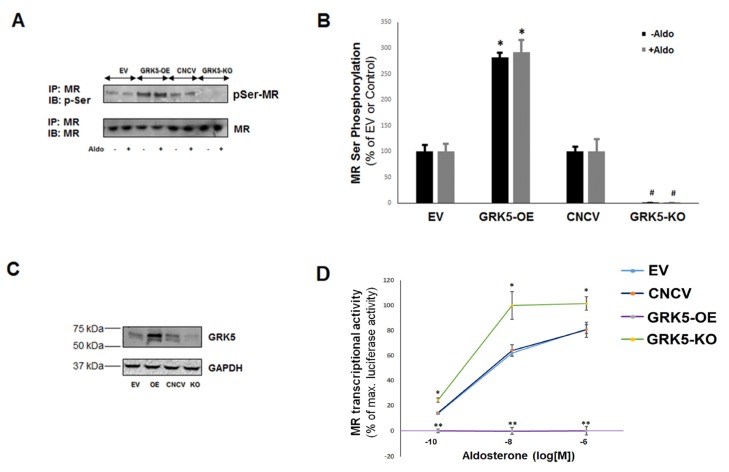
GRK5 phosphorylates and inhibits the MR in cardiac myocytes. (**A**) Western blotting for phosphoserine content of immunoprecipitated MR in response to 100 nM aldosterone (Aldo) stimulation for 15 min or vehicle (DMSO) in GRK5-overexpressing (GRK5-OE) or control, mock lentivirus-infected (EV) H9c2 myocyte lysates or in lysates from myocytes that had GRK5 genetically deleted (GRK5-KO) and their respective control cells (CNCV, CRISPR (Clustered regularly interspaced short palindromic repeats) negative control virus). IP: immunoprecipitation; IB: Immunoblotting. Representative blots are shown in (**A**) and the relative densitometric quantitation of five independent experiments performed in duplicate is shown in (**B**). * *p* < 0.05, vs. EV; ^#^
*p* < 0.05, vs. CNCV; *n* = 5. (**C**) Immunoblotting for GRK5 in extracts from H9c2 cardiomyocytes, transfected with empty vector/mock lentivirus (EV), full-length wild-type GRK5-encoding lentivirus to overexpress GRK5 (OE), CRISPR negative control (mock) lentivirus (CNCV), or CRISPR rat GRK5-specific lentivirus to knockout GRK5 (KO). Total protein extracts were prepared 48 h post-infection and then separated on a 4%–20% SDS-PAGE gel. A representative blot is shown, including glyceraldehyde 3-phosphate dehydrogenase (GAPDH) as loading control, of five independent experiments performed in duplicate, confirming GRK5 overexpression and deletion in OE and KO cells, respectively. (**D**) Transcriptional activity of the MR in response to various concentrations of Aldo in H9c2 cardiomyocytes overexpressing GRK5 (GRK5-OE) or having GRK5 genetically deleted (GRK5-KO). EV: Empty vector; CNCV: CRISPR negative control virus; * *p* < 0.05, between GRK5-KO and EV or CNCV; ** *p* < 0.05, between GRK5-OE and EV or CNCV; *n* = 5 independent measurements with triplicate samples per Aldo concentration.

**Figure 4 ijms-21-02868-f004:**
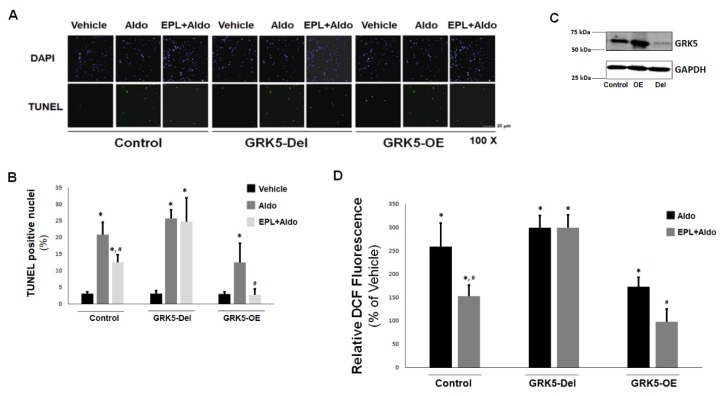
Effects of GRK5 in Aldo-induced apoptosis in adult rat ventricular myocytes (ARVMs): (**A**,**B**) Apoptotic cell death, measured by terminal deoxynucleotidyl transferase dUTP nick-end labeling (TUNEL), in cultured ARVMs having GRK5 genetically (via CRISPR) deleted (GRK5-Del) or overexpressing GRK5 (GRK5-OE) and treated with 100 nM Aldo or 100 nM Aldo in the presence of 10 µM eplerenone (EPL + Aldo) for 24 hrs. Representative images of TUNEL-positive nuclei identified by 4′,6-diamidino-2-phenylindole (DAPI) counterstaining are shown in (**A**), and the quantitation of the TUNEL imaging results are shown in (**B**). Control: CRISPR-negative control virus (CNCV)-infected cells. * *p* < 0.05, vs. vehicle; ^#^, *p* < 0.05, vs. Aldo; *n* = 5 independent experiments per transfection/treatment. (**C**) Immunoblotting for GRK5 in extracts from cultured ARVMs, transfected with control empty vector/mock lentivirus (Control), full-length wild-type GRK5-encoding lentivirus to overexpress GRK5 (OE), or CRISPR rat GRK5-specific lentivirus to delete the gene for GRK5 (Del). Total protein extracts were prepared 48 h postinfection and then separated on a 4%–20% SDS-PAGE gel. A representative blot is shown, including GAPDH as loading control, of five independent experiments performed in duplicate, confirming GRK5 overexpression and deletion in OE and Del ARVMs, respectively. (**D**) Reactive oxygen species (ROS) generation, as measured with a 2′,7′-dichlorofluorescein diacetate (DCFDA)-based assay kit, in cultured ARVMs having GRK5 genetically (CRISPR-mediated) deleted (GRK5-Del) or overexpressing GRK5 (GRK5-OE) and treated with 100 nM aldosterone (Aldo) or 100 nM aldosterone in the presence of 10 µM eplerenone (EPL + Aldo) for 24 hrs. Results are expressed as % of the fluorescence measured upon vehicle (DMSO) treatment for each cell clone. Control: Empty vector/mock lentivirus-infected ARVMs. Eplerenone alone had no effect. * *p* < 0.05, vs. vehicle; ^#^
*p* < 0.05, vs. Aldo; *n* = 6 independent experiments per transfection/treatment.

**Figure 5 ijms-21-02868-f005:**
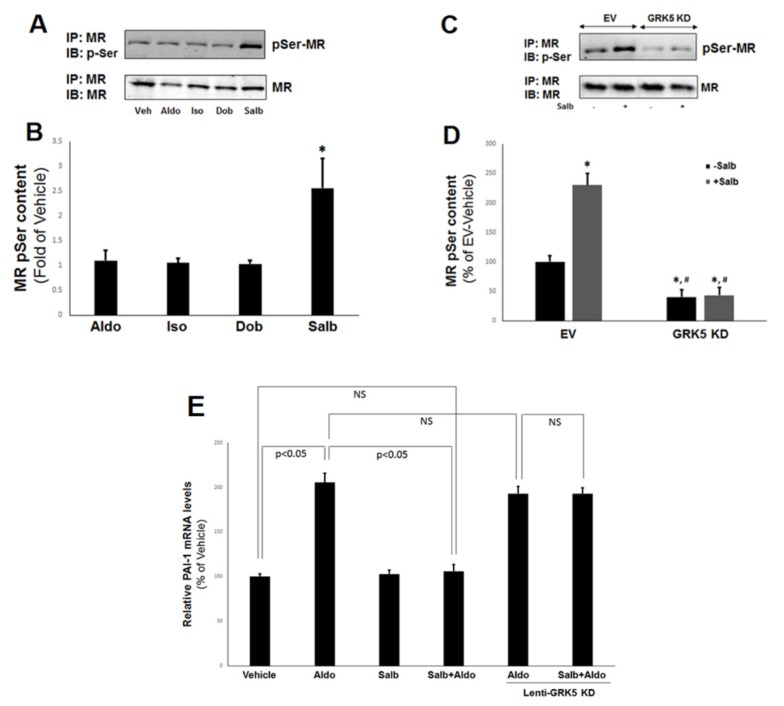
β_2_AR stimulates the GRK5-dependent phosphorylation and inhibition of the cardiac MR. (**A**,**B**) Immunoblotting for phosphoserine content of the MR in response to 15-min-long treatments with 100 nM Aldo, 10 µM isoproterenol (Iso), 1 µM dobutamine (Dob), or 10 µM salbutamol (Salb) in native ARVM lysates. Representative blots are shown in (**A**), and the relative densitometric quantitation of five independent experiments done in duplicate is shown in (**B**). * *p* < 0.05, vs. any other treatment; *n* = 5. (**C**,**D**) Immunoblotting for phosphoserine content of the MR in response to 10 µM salbutamol (Salb) for 15 min or vehicle (DMSO) in ARVMs transfected with a GRK5 kinase-dead dominant negative mutant (GRK5 KD) lentivirus or control, mock (empty vector, EV) lentivirus. Representative blots are shown in (**C**), and the relative densitometric quantitation of six independent experiments done in duplicate is shown in (**D**). * *p* < 0.05, vs. EV/−Salb; ^#^, *p* < 0.05, vs. EV/+Salb; *n* = 6. IP: immunoprecipitation; IB: Immunoblotting. (**E**) mRNA levels of plasminogen activator inhibitor (PAI)-1 in response to a 2-hr-long treatment of 100 nM Aldo alone or in the presence of 10 µM salbutamol (Salb + Aldo) in ARVMs. Lenti-GRK5 KD: Cells transfected with a GRK5 kinase-dead mutant lentivirus. 18S rRNA levels were used for normalization of the results. NS: Not significant at *p* = 0.05; one-way ANOVA with Bonferroni test; *n* = 6 independent experiments done in duplicate samples/condition.

**Figure 6 ijms-21-02868-f006:**
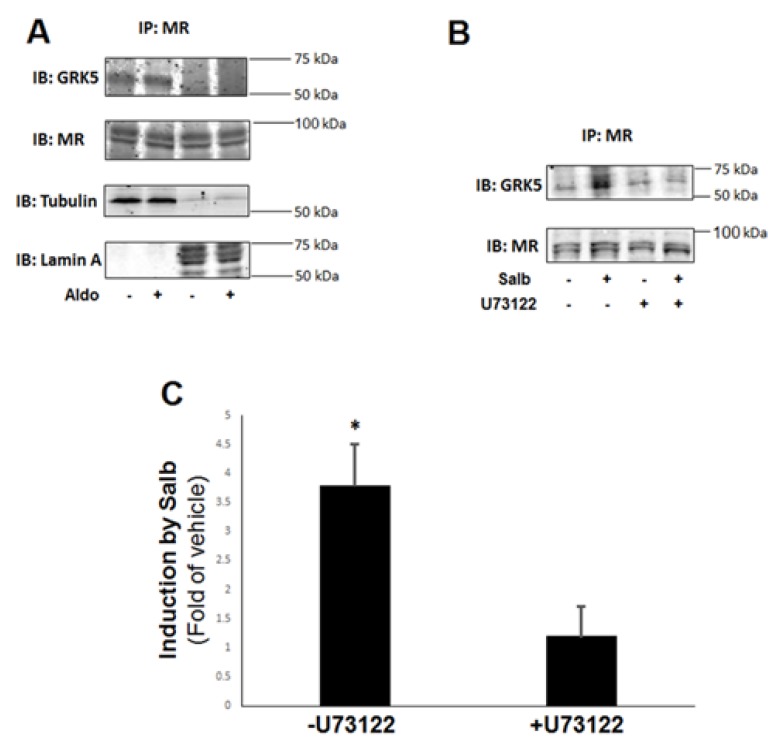
β_2_AR induces GRK5 translocation to the cytoplasm for subsequent MR phosphorylation. (**A**) H9c2 cardiomyocytes were untreated or treated with 100 nM Aldo for 30 min. Subsequently, whole cell protein lysates were prepared and subjected to subcellular fractionation. Cytosolic and nuclear extracts fractions were separately immunoprecipitated (IP) with an anti-MR antibody, and the IPs were then immunoblotted (IB) for GRK5. Representative blots from six independent experiments are shown, including blots for MR to confirm equal amounts immunoprecipitated and for tubulin (cytosolic marker) and lamin A (nuclear marker) to confirm the identity of the lysate fraction (cytoplasm or nucleus, respectively). No GRK5 immunoreactivity could be detected in the MR IPs derived from the nuclear lysates. (**B**,**C**) H9c2 cardiomyocytes were treated with 10 µM salbutamol (Salb) for 30 min in the presence or absence of 10 µM U73122, and then, whole cell lysates were prepared to IP the MR, followed by IB for GRK5. Representative blots are shown in (**B**), including blots for MR to confirm equal amounts immunoprecipitated, and the relative densitometric quantitation of six independent experiments done in duplicate is shown in (**C**). * *p* < 0.05; *n* = 6.

**Figure 7 ijms-21-02868-f007:**
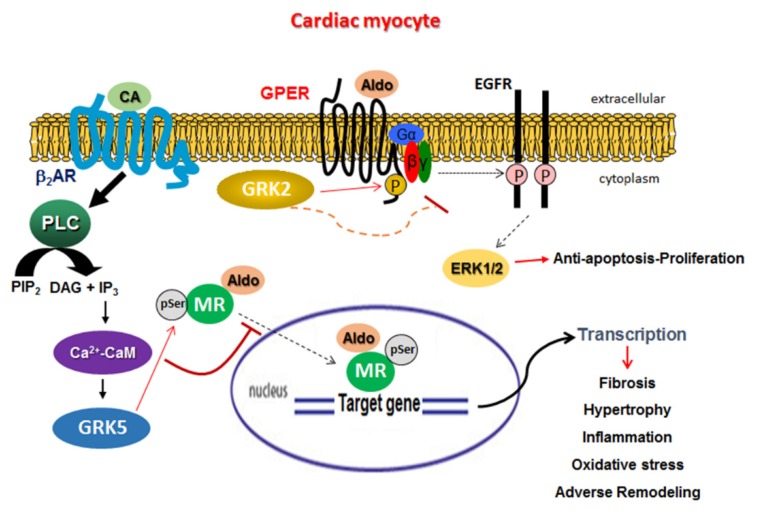
Schematic illustration of the signaling interplay between GRK5 and MR as well as between GRK2 and GPER inside cardiac myocytes. GRK5, activated by the cardiac β_2_AR via PLCβ-Ca^2+^-CaM signaling, translocates to the cytoplasm to phosphorylate the MR and inhibit its transcriptional activity as a result of the Aldo–MR complex’s cytoplasmic-nuclear shuttling blockade. GRK2, on the other hand, phosphorylates and desensitizes the agonist-activated GPER at the plasma membrane. Aldo: Aldosterone; CA: Catecholamine; P: Phosphorylation; pSer: Phosphorylated serine; PLC: Phospholipase C; PIP_2_: Phosphatidylinositol 4,5-bisphosphate; DAG: Diacylglycerol; IP_3_: Inositol 1,4,5-trisphosphate; CaM: Calmodulin; ERK: Extracellular signal-regulated kinase. See text for more details and for all other molecular acronyms’ descriptions. Solid arrows denote direct activation/action, dotted arrows multi-step activation or translocation, and **⊣** denotes inhibition.

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
