# Peer review of "Antagonistic Roles of GRK2 and GRK5 in Cardiac Aldosterone Signaling Reveal GRK5-Mediated Cardioprotection via Mineralocorticoid Receptor Inhibition"

_ijms, 2020, doi:10.3390/ijms21082868_

Round 1
Reviewer 1 Report
The authors have performed experiments to investigate the roles of GRK5 and GRK2 in the regulation of MR activity in cardiomyocytes. Overall, this study is very interesting. The authors carefully performed the experiments and manuscript contains some novel information. However, there are several relatively minor but important concerns that should be easily addressed to strength the manuscript. The specific points are below.
Critiques:
- This reviewer does not understand why the authors used H9c2 and ARVM cells for each experiment. Please include the explanation in the methods section. In this regard, please also show the data regarding basal MR expression in both cells.
- It is very difficult to see the Figure 3D. Please replace better figure.
- Additional experiments should be performed to examine the effect of eplerenone on aldosterone-induced MR transcriptional activity in GRK-OE and -KO cells (Figure 3D). These measurements should be particularly important for understanding the relationship contribution of GRK5 to changes in MR activity in response to aldosterone and eplerenone.
- It is also difficult to see Figure 4A. Please increase its size.
Author Response
We thank this reviewer for his/her overall positive and kind remarks about the quality of our work.
- This reviewer does not understand why the authors used H9c2 and ARVM cells for each experiment. Please include the explanation in the methods section. In this regard, please also show the data regarding basal MR expression in both cells.
Author response: H9c2 cells were used because they are a universally accepted and widely used cell model system for signaling studies in cardiac cells/myocytes. Nevertheless, they are not bona fide cardiac myocytes (for instance, they do not contract); for this reason, ARVMs, which are bona fide, fully differentiated cardiac myocytes, were also used. This is now clearly stated in "Materials and Methods", lines 1-5, p. 12, of the revised manuscript, as per the reviewer`s request.
As for the basal MR expression, we do not have any data on that, unfortunately. However, and as stated in the text (lines 25-26 of p. 2 of the revised manuscript), both H9c2 cells and ARVMs are known to express the MR endogenously (e.g. see Refs. 13 & 14 of the revised manuscript and Rude et al. Hypertension. 2005;46(3):555-61).
- It is very difficult to see the Figure 3D. Please replace better figure.
Author response: Done. Apologies for the bad resolution of the original figure.
- Additional experiments should be performed to examine the effect of eplerenone on aldosterone-induced MR transcriptional activity in GRK-OE and -KO cells (Figure 3D). These measurements should be particularly important for understanding the relationship contribution of GRK5 to changes in MR activity in response to aldosterone and eplerenone.
Author response: We thank the reviewer for this very pertinent suggestion. However, performing these additional experiments will take a significant amount of time (even more so now, during the current COVID-19 pandemic crisis). On the other hand, we do not feel these experiments are necessary, as they will not provide additional essential information. The experiments of Fig. 3D were done to merely show the effect of GRK5 alone on MR transcriptional activity. Investigation of GRK5`s potentially additive or synergistic effects with eplerenone on MR inhibition are beyond the scope of the present study. Besides, our data in ARVMs (Fig. 4) directly hint at the effects of the GRK5-eplerenone combination on cardiac MR activity. We hope that the reviewer understands and agrees on this.
- It is also difficult to see Figure 4A. Please increase its size.
Author response: Done. Again, apologies for the bad resolution of the original figure.
Reviewer 2 Report
The authors titled the manuscript: Differential Roles of GRK2 and GRK5 in Cardiac Aldosterone Signaling Suggest GRK5-Mediated Cardio-protection Via Mineralocorticoid Receptor Inhibition. but the GRK2 role was not supported with results. the heading of the paper is very long and very vague!
in Materials, the font size was all over the place.
The quality of the figures is a bit poor, the scales are not readable and are shown in grey.
In Figure 5 they decided to compare the bar graphs with lines, yet still, salb bar was standing alone, without any comparison.
The results are a bit poorly written and discussion is not fully supporting the results.
Author Response
We thank this reviewer for his/her kind and positive words about the quality of our work.
The authors titled the manuscript: Differential Roles of GRK2 and GRK5 in Cardiac Aldosterone Signaling Suggest GRK5-Mediated Cardio-protection Via Mineralocorticoid Receptor Inhibition. but the GRK2 role was not supported with results. the heading of the paper is very long and very vague!
Author response: We respectfully disagree. We believe we have sufficiently documented the detrimental role of GRK2 in aldosterone signaling via its effects on GPER. Besides, that has been already documented in other studies, as well (see Refs. 23 & 30 of the revised manuscript for relevant reviews).
As for the length of the title, we have rephrased it according also to the editor`s suggestion. We used a lengthy title exactly so that it would be clear and not vague at all!
in Materials, the font size was all over the place.
Author response: Corrected. Apologies for the inconvenience.
The quality of the figures is a bit poor, the scales are not readable and are shown in grey.
Author response: Corrected. Apologies for the inconvenience.
In Figure 5 they decided to compare the bar graphs with lines, yet still, salb bar was standing alone, without any comparison.
Author response: That is because "Salb" alone had no effect (i.e. it behaved essentially as vehicle). We hope the reviewer understands now the reason behind this.
The results are a bit poorly written and discussion is not fully supporting the results.
Author response: We have significantly revised the text of both "Results" and "Discussion". We hope this now satisfies this reviewer.
Round 2
Reviewer 1 Report
- Additional experiments should be performed to examine the effect of eplerenone on aldosterone-induced MR transcriptional activity in GRK-OE and -KO cells (Figure 3D). These measurements should be particularly important for understanding the relationship contribution of GRK5 to changes in MR activity in response to aldosterone and eplerenone.
Author response: We thank the reviewer for this very pertinent suggestion. However, performing these additional experiments will take a significant amount of time (even more so now, during the current COVID-19 pandemic crisis). On the other hand, we do not feel these experiments are necessary, as they will not provide additional essential information. The experiments of Fig. 3D were done to merely show the effect of GRK5 alone on MR transcriptional activity. Investigation of GRK5`s potentially additive or synergistic effects with eplerenone on MR inhibition are beyond the scope of the present study. Besides, our data in ARVMs (Fig. 4) directly hint at the effects of the GRK5-eplerenone combination on cardiac MR activity. We hope that the reviewer understands and agrees on this.
I disagree that these additional experiments are not so important, because your present data merely provide a hint, but has less supportive data. However, I really understand your serious Corona Virus situation. Therefore, the authors can explain these issues as a weak point of this study and/or future direction.
Author Response
We thank this reviewer for his/her understanding. We never suggested these experiments were not important; on the contrary, every additional piece of experimental information one can get is always important! However, given the current situation (our lab is currently closed due to the pandemic), we appreciate the reviewer`s leniency and leeway here.
However, since we indeed plan to perform these experiments in a future study, we have included a statement in the "Discussion" of the newly revised manuscript, as follows: "Moreover, it appears essential for the cardio-protection afforded by MRA drugs like eplerenone against Aldo`s deleterious effects. Nevertheless, the precise effects of GRK5-dependent MR phosphorylation on eplerenone’s MR inhibitory efficacy warrant further investigation and will be the focus of our future studies. In addition, identification of the exact Ser/Thr residues of the MR phosphorylated by GRK5 is already under way in our laboratory"; lines 1-5, p. 11 (highlighted in yellow). We hope this now satisfies this reviewer and we thank him/her again for their understanding.
Reviewer 2 Report
Dear Authors,
There are so many programs available for high-quality figure preparation. Please use a suitable programme to present your data next time!
Author Response
Thank you for the kind suggestion. We will oblige next time!